# In Vitro and In Vivo Biocontrol of Tomato Fusarium Wilt by Extracts from Brown, Red, and Green Macroalgae

**Yasser S. Mostafa [1], Saad A. Alamri [1], Sulaiman A. Alrumman [1], Mohamed Hashem [1,2], Mostafa A. Taher [1,3] and Zakaria A. Baka [4,\*]**

1   Department of Biology, College of Science, King Khalid University, Abha P.O. Box 9004, Saudi Arabia; ysmosutafa@kku.edu.sa (Y.S.M.); saralomari@kku.edu.sa (S.A.A.); salrumman@kku.edu.sa (S.A.A.); mhashem@kku.edu.sa (M.H.); mtaher@kku.edu.sa (M.A.T.)
2   Department of Botany and Microbiology, Faculty of Science, Assiut University, Assiut P.O. Box 71515, Egypt
3   Department of Botany, Faculty of Science, Aswan University, Aswan P.O. Box 81528, Egypt
4   Department of Botany and Microbiology, Faculty of Science, Damietta University, New Damietta P.O. Box 34517, Egypt
\*   Correspondence: zakariabaka@du.edu.eg

**Abstract:** Fusarium wilt caused by *Fusarium oxysporum* f. sp. *lycopersici* (FOL) in tomatoes is globally recognized as one of the most significant tomato diseases, both in fields and in greenhouses. Macroalgae contain a diversity of bioactive complexes. This research was carried out to assess the value of the extracts from three macroalgae (*Sargassum dentifolium* belongs to Phaeophyta, *Gracilaria compressa* belongs to Rhodophyta, and *Ulva lactuca* belongs to Chlorophyta) against wilt disease in tomato plants. The fungal pathogen was isolated from diseased tomato plants growing in several parts of Saudi Arabia. Isolates of the pathogen were identified by morphological and molecular methods. Three organic solvents, in addition to water, were used for extraction to assess the effect of reducing FOL hyphal growth on potato dextrose agar (PDA). Radial reductions in pathogen hyphal growth were seen with all of the solvent and water extracts, but the three macroalgae methanol extracts that were tested showed the greatest reduction in pathogen hyphal growth. The total phenol content of the *S. dentifolium* extract was higher than that of the other two macroalgae. The phenolic compounds showed variability in all of the extracts that were identified and calculated by high-performance liquid chromatography (HPLC). Phloroglucinol (7.34 mg/g dry weight), vanillic acid (7.28 mg/g dry weight), and gallic acid (6.89 mg/g dry weight) were the phenolic compounds with the highest concentrations in the *S. dentifolium*, *G. compressa*, and *U. lactuca* extracts, respectively. The mycelium of FOL treated with a crude macroalgae extract of tested at 100 μg/mL was examined with a scanning electron microscope. The results showed an obvious difference between the extract-treated and untreated hyphae. The extract-treated hyphae collapsed and bruised, as well as; empty and dead. In the greenhouse experiment, *S. dentifolium* powder was used to evaluate its effect on disease decline. It led to a decrease in disease severity of 40.8%. The highest total yield (560.8 g) was obtained from the plants treated with *S. dentifolium* powder. We recommend the use of macroalgae extracts to combat fungal phytopathogens. Because chemical fungicides are extremely toxic to humans and the environment, macroalgae extracts are a good alternative that can be widely and safely used in the field.

**Keywords:** biocontrol; *Fusarium oxysporum lycopersici*; *Gracilaria compressa*; phenolic compounds; *Sargassum dentifolium*; scanning electron microscopy; tomato wilt; *Ulva lactuca*

## 1. Introduction

Tomatoes (*Lycopersicon esculentum* Mill.) are significant organic plants in Saudi Arabia, with a production per year of about 306,000 tones [1,2]. Tomato wilt, a catastrophic disorder caused by *Fusarium oxysporum* f. sp. *lycopersici* (FOL) drastically decreases tomato yield in both greenhouse and field conditions [3]. Under optimal infection conditions, yield losses

can reach up to 90% [4]. Fighting tomato wilting with standard chemical methods is a challenge. In addition, this type of management is limited in organic farming [5]. Moreover, the control of this disease depends on the application of some chemical fungicides in the late stages of tomato production [6].

Biocontrol of plant diseases is an efficient means of minimizing the unsafe effects of chemical fungicides, having been declared to improve safety levels and minimize environmental impacts [7]. Large algae are marine organisms that are regarded as the main source of bioactive compounds [8]. There are plentiful accounts of macroalgae compounds with a broad range of biological activities, including antifungal and antibacterial [9,10].

Macroalgae mostly consisted of diverse populations of green, brown, and red large algae. They are able to improve the chemical and physical characteristics of the soil. Through the application of macroalgae extracts, plant diseases can be reduced and plant development encouraged.

Kausalya and Rao [11] pointed out that the presence of antifungal agents in the extraction of seaweeds, for example, *Sargassum polycystum* and *S. tenerrimum* may be used to produce innovative agents for the benefit of humankind. Martin [12] tested extracts from large algae on plants and found a significant decrease in the occurrence of strawberry and turnip diseases caused by *Botrytis cinerea* and Erysiphe polygoni, respectively, owing to the occurrence of a category of biologically effective compounds created by macroalgae. Furthermore, Hellio et al. [13] examined the antimicrobial effects of many algal extracts and noted a marked reduction in the growth of the tested fungi.

Yi et al. [14] used different organic solvents to extract antibiotics from the extracts of green, brown, and red macroalgal species. They found that these extracts had the strongest antifungal activity. Khanzada et al. [15] examined many constituents of ethanol extract from *Solieria robusta* (Rhodophyta) to examine their antifungal action in fruits and discovered that all of the concentrations prevented fungal development. The aqueous extract had the highest effect, followed by chloroform, methanol, ethanol, and ethyl acetate. In addition, the highest inhibitory effects were seen against *Candida albicans*, *Mucor ramaniannus*, and *Aspergillus niger*, when the methanol extracts of *Padina pavonica* (brown algae) and *Rhodomela confervoides* (red algae) were used. Perez et al. [16] reported that 10 algal extracts significantly suppressed the growth of *Colletotrichum muiradnegeli*, but not *Aspergillus flavus*.

The benefit of using macroalgae extracts, as a fertilizer is that it is quickly absorbed by plants within hours of use and is safe for humans, animals, and the environment [17]. Thus, macroalgae may play an important function in agriculture because they reduce the application of pesticides and fertilizers. Consequently, the usage of algae and their extracts provides higher agricultural production.

The main reason for this investigation was to evaluate the possibility of controlling tomato Fusarium wilt by using extracts derived from three marine macroalgae in vitro and under greenhouse conditions. In addition, measurements of phenolic compounds in these macroalgae extracts were also considered.

## 2. Materials and Methods

### 2.1. Isolation of Fusarium from Infected Tomato Plants and Pathogenicity Testing

The Fusarium pathogen in tomatoes was isolated using the techniques of Tsegaye and Tesfaye [18]. The pathogenicity testing of Fusarium isolates was established by Koch's principle. Tomato seeds of the same size (Susceptible variety Super Strain B) were sterilized with 2% NaOCl for 5 min and cleaned three times with distilled water. Afterward, the pathogen inoculum was designed by culturing each isolate in a PDA medium at 28 °C for 14 days. Then, 10 mL of sterilized distilled water were added to each Petri dish, and the colonies were scraped with a sanitized needle. The conidial suspension of each isolate was calibrated to $6 \times 10^6$/mL and we inoculated 25 tomato plants using an aerosol. Following inoculation, the plants were kept in plastic bags for 48 h to maintain high moisture. Finally, after 48 h, the plastic bags were removed and the tomato plants were potted and protected

in greenhouse conditions. After two weeks, the results were evaluated; the experiment was repeated more.

### 2.2. Pathogen Identification

#### 2.2.1. Characteristics of Culture and Morphology of Fusarium Isolates

The characteristics of Fusarium isolates growing on PDA nutrient medium were investigated by culturing every isolate three times at 25 °C. Seven days after inoculation, the characteristics of the culture such as colony form, edge shape, color, reverse color, pigmentations, and centrifugal growth were tested and growth rates were measured. Moreover, the growth degree value was estimated by deducting the centrifugal growth value on the first day from the centrifugal growth value on the next day for seven days. Lastly, the average of entire variances was calculated to show the growth degree of this particular isolate [19].

Slide cultures were created for each isolate by removing 5 mm mycelial fragments from the pure culture made seven-days-prior and inoculating them in a PDA medium. Moreover, they were positioned on sterile glass slides and overlaid with a coverslip. Then, the slides were positioned in a sterile Petri dish with dampened filter paper and incubated at 25 °C [20]. Afterward, the mycelium produced in the slide culture was microscopically examined and morphologically differentiated at the species level at a magnification of 40× using a microscope (Olympus Type, Carl Zeiss Microscopy, Oberkochen, Germany).

#### 2.2.2. Identification of Fusarium Isolates by Molecular Technique

#### DNA Extraction

DNA was extracted with cetyltrimethylammonium bromide (CTAB). Isolates of the pathogen were cultured at 25 °C for seven days using a PDA medium. The mycelium was then, moved to antiseptic 1.5-mL Eppendorf microtubes and ground with liquid nitrogen to make a uniform powder. Afterward, 700 μL CTAB buffer (100 mM Tris-HCl, 1.0 M NaCl, 10 mM EDTA, pH 8.0) including CTAB (2%, *w/v*) was inserted to 100 g ground powder. After the addition, the microtubes were kept at 65 °C for 45 min. and centrifuged at 10,000× *g* for 10 min. A 650 μL supernatant mixture was centrifuged at 10,000× *g* with identical volumes of isoamyl alcohol and chloroform for 10 min. at 25 °C. The supernatant was separated, 0.7 mL of cold isopropyl alcohol was added to the combination, and the mixture was deposited at −20 °C for 20 min. Then, the tubes were centrifuged at 10,000× *g* for 5 min. The DNA-enclosing precipitate was washed twice with 70% ethyl alcohol and centrifuged at 10,000× *g* for 5 min. each time. Afterward, the pellet was dried in air and the ready DNA was resuspended in 30 μL TE buffer (10 mM Tris-HCl, pH 8.1 mM EDTA). Lastly, the nucleic acid concentration was checked with a NanoDrop 1000 Spectrophotometer (Thermo Fisher Scientific, Wilmington, DE, USA). The consistency of every DNA sample was verified on a 12 g $L^{-1}$ agarose gel. The properties of DNA obtained on a 1% agarose gel using a DAC gel using electrophoresis and DNA nontoxic-staining were considered.

#### Polymerase Chain Reaction (PCR)

Clarification of the ITS1 locality in PCR was attained by two primers: ITS1 (5′-TCCGTAGGTGAACCTGCGG-3′) and ITS4 (5′-TCCTCCGCTTATT GATATGC-3′). The PCR reaction was performed on a Bio-Rad thermal cycler (S1000™) (Shanghai, China). PCR is unique in that it has 40 cycles of 94 °C for 5 min., 94 °C for 1 min., primer curing at 53 °C for 45 s, and primer expansion at 72 °C for 90 s with primary denaturation at 94 °C for 5 min and 10 min final elongation at 72 °C. Afterward, the reaction was performed in 25 μL containing a 2 μL DNA template, 12.5 μL master mix, and 10 pmol of each primer. PCR results were electrophoresed at 80 V for 1 h on a 0.8 agarose gel in Tris-borate-EDTA buffer at pH 8. The gels were stained with DNA—safe-stain (10 mg/mL) and monitored in a gel certification system (Alpha Innotech, San Leandro, CA, USA). Furthermore, the product sequence taken from PCR in the ITS1 area was determined. Nucleotide sequences gained

by local BLAST (http://blast.ncbi.nlm.nih.gov/Blast.cgi) (15 October 2021) were tested and contrasted to GenBank parallel sequences.

### 2.3. Sampling of Macroalgae

Three macroalgae were collected on the coasts of Jazan (Saudi Arabia) and Hurghada (Egypt). The algae were quickly collected and carefully washed with fresh running water to eliminate foreign material. Next, the macroalgae were drained and cleaned with absorbent paper then dried in air at 45 °C for five days. The macroalgae were completely dried and minced in a motorized mill. As stated by Kumar et al. [21], the macroalgae have been identified as *Ulva lactuca* (green algae), *Sargassum dentifolium* (brown algae), and *Gracilaria compressa* (red algae).

### 2.4. Preparation of Algal Extracts

Three organic solvents (chloroform, acetone, and methanol) were used besides water, and 200 mL of every solvent was added to 50 g ($v/w$) of macroalgal powder. Then, the mixture was shaken in a shaker at room temperature for 10 days. Afterward, the extract was filtered through a cheesecloth, followed by Whatman No. 2 filter paper [22].

### 2.5. Antifungal Activity of Algal Extracts In Vitro

For assessing the antifungal activity of macroalgae extracts, 20, 40, 60, 80, and 100 µg/mL. of each solvent were dispensed into sterilized PDA plates. PDA plates are amended with each of the solvents to prepare the algae extracts; chloroform, acetone, and methanol were used as controls. A 5-mm-diameter circle was then removed from the edges of the vigorously developing fungal colonies and moved to the middle of the medium. In addition, all inoculated plates were stored in an incubator at 25 °C and tested every day until the top of the medium was completely coated with fungus. Next, the radial growth of the fungal pathogen was measured. The hyphal growth inhibitory effect of the extract was determined for each concentration by means of the following formula: % inhibition = $(dc - dt)/dc \times 100$, where $dc$ is the average increase in control hyphal growth and $dt$ is the average increase in growth in treated mycelium [23].

### 2.6. Determination of Total Phenolic Content

The method of Sultana et al. [24] was used to determine the total phenols in the extracts of tested macroalgae. Five milligrams of ethanol extract from each macroalga were dissolved in 2 mL of 96% ethyl alcohol, and then 5 mL of distilled water and 0.5 mL of 50% Folin–Ciocalteu reagent were added. The mixtures were incubated for 5 min and then 1 mL of 5% sodium carbonate was added. The solutions were combined and incubated in the dark for 1 h. The absorbance was calculated three times at a wavelength of 725 nm by means of a UV-Vis spectrophotometer. Gallic acid with a series of concentrations (0, 5, 15, and 20 ppm) was used as the standard. The phenolic content in the sample was determined by a regression equation using a gallic acid standardization curve and conveyed as mg gallic acid/g extract.

### 2.7. Determination of Phenolic Compounds by High-Performance Liquid Chromatography (HPLC)

The phenolic components of the methanol extract of the tested macroalgae were identified using the Agilent 1200 Series HPLC scheme (Santa Clara, CA, USA). This consists of a reverse-phase C18 column (150–4.6 mm, unit size 0.5 mm), a quarterly pump, and a UV sensor placed at 330 nm. Two-step elution was carried out at a run speed of 1 mL/min. The movable stage was formed of acetonitrile (A) and methanol (B). In addition, the splitting of phenolic composites started with a linear slope of 20% B (0–10 min), 40% B (10–25 min), 90% B (25–30 min). The temperature of the column was maintained at 35 °C and the insertion volume was 20 µL. Moreover, the identification of phenolic constituents was completed depending on retention time and UV spectrum using present standards. The

number of phenolic constituents was completed by a peak zone validated at 330 nm. The outcomes were registered as mg/g algal powder.

### 2.8. Scanning Electron Microscopy (SEM)

The macroalgae extract in methanol (100 g/mL) displayed the maximum antifungal activity of FOL; consequently, SEM was used to investigate its culture. In the case of the control, PDA plates were amended with methanol; alone. In the SEM technique, fungal masses (1 mm$^3$) were prepared with 3% glutaraldehyde in a 0.1 M sodium cacodylate buffer at pH 6.8, dehydrated with an orderly acetone sequence, and subjected to a critical point dryer (Polaron CPD 7501, Watford, UK). Afterward, the samples were covered with gold-palladium using the JFC1600 Auto Fine Coater (Polaron CPD 7620). Fungal mycelium was then observed and photographed under JEOL SEM6400JSM-6360LV (JEOL, Ltd., Akishima, Japan).

### 2.9. Greenhouse Experiment

Because S. dentifolium extract had the strongest inhibitory effect on FOL hyphal growth, this algal powder was deemed as a biological control agent for FOL and used in a greenhouse experiment. In this study, autoclaved sterile sandy-loam soil was infested with an inoculum of sorghum grains of 3% (*w/w*)/kg soil (2–6% colonization of FOL on sorghum grains used as a bait and 3000 CFU g$^{-1}$ of soil). These infested sorghum grains (autoclaved before infestation) were provided by the Microbiology Laboratory of Damietta University, New Damietta, Egypt). Infested pots were irrigated and set aside for seven days to guarantee pathogen dispersal in the soil before seed sowing. The infested soil was amended with dry algal powder in the ratio 1 g algal powder: 1 kg soil *w/w* at seed sowing. Ten seeds were sown in each pot; four replicates were used for each treatment. The control treatment was carried out with a chemical fungicide (benomyl) as seed coating (3 g benomyl/kg) seeds before seed sowing. In this experiment, four treatments were evaluated; infected control, algal powder only, fungicide only, and algal powder + fungicide. Pots were kept in the greenhouse until the end of the experiment (duration: 60 days), and the tomato yield was weighted for all treatments. Seeds of tomatoes were obtained from the Agricultural Research Center, Cairo, Egypt. Seeds were surface-sterilized in a 2% sodium hypochlorite solution for 2 min before sowing [25].

### 2.10. Effect of Algal Extract of S. dentifolium on Disease Assessment and Fruit Yield

Wilt symptoms that emerged after seven days after inoculation were recorded as disease severity and measured on a 1–5 scale [26]; 0 = plants without spots, 1 = a leaf with a few spots, 2 = more spots increased and scattered, 3 = spots combined and became larger, 4 = the necrotic spots reached half of the leaf, 5 = the majority of the leaf was necrotic and dropped. Disease severity was evaluated by the following formula: disease severity % = (n × v)/9 N × 100, where: (n) = number of plants in each category. (v) = numerical values of symptom category, (N) = total number of plants. (9) = maximum numerical value of symptom category. Furthermore, disease reduction was measured by the following formula: Disease reduction % = disease severity in control–disease severity in treatment/disease severity in control × 100. Finally, the weight of tomato fruits per plant was determined after the application of algal extract. Three harvests as replicates were considered.

### 2.11. Data Analysis

Software System, version 13 (StatSoft Inc., Tulsa, OK, USA) was utilized for the data analysis. Pearson's correlation coefficient and a principal component analysis were used to verify the relationships between variables. All data are expressed as the mean ± standard deviation. The significance level was set to $p < 0.05$.

## 3. Results

### 3.1. Characteristics of Culture and Morphology of Fusarium Isolates

Ten Fusarium isolates were obtained from tomato plants showing wilt. The isolates displayed purple or whitish-purple colony colors. On the backside, the isolates exhibited purple pigmentation. Two of the isolates were evaluated as quick growing (F1 and F3) and the remainder had an average growth rate (F2, F4–F10). Generally, macroconidia were more or less plentiful, 3–5 septa, straight, sometimes positioned in bunches, curved at both sides, a few fusiform in shape, and with more ventral curving than the opposite side. Most of the microconidia were elongated, septated with long or short phialides some were sphere-shaped.

### 3.2. Growth Rate and Radial Growth of FOL Isolates

Isolates F 1 and F 2 showed the highest growth rates on a PDA nutrient medium at 25 °C 2 and 15 days after incubation. F1 recorded 7.28 and 7.42 mm, respectively, while F2 displayed 52.12 and 55.19 mm, respectively. Isolates F1 and F3 were classified as quick growing and the other isolates had an average growth rate (F2, F4–F10) (Figure 1).

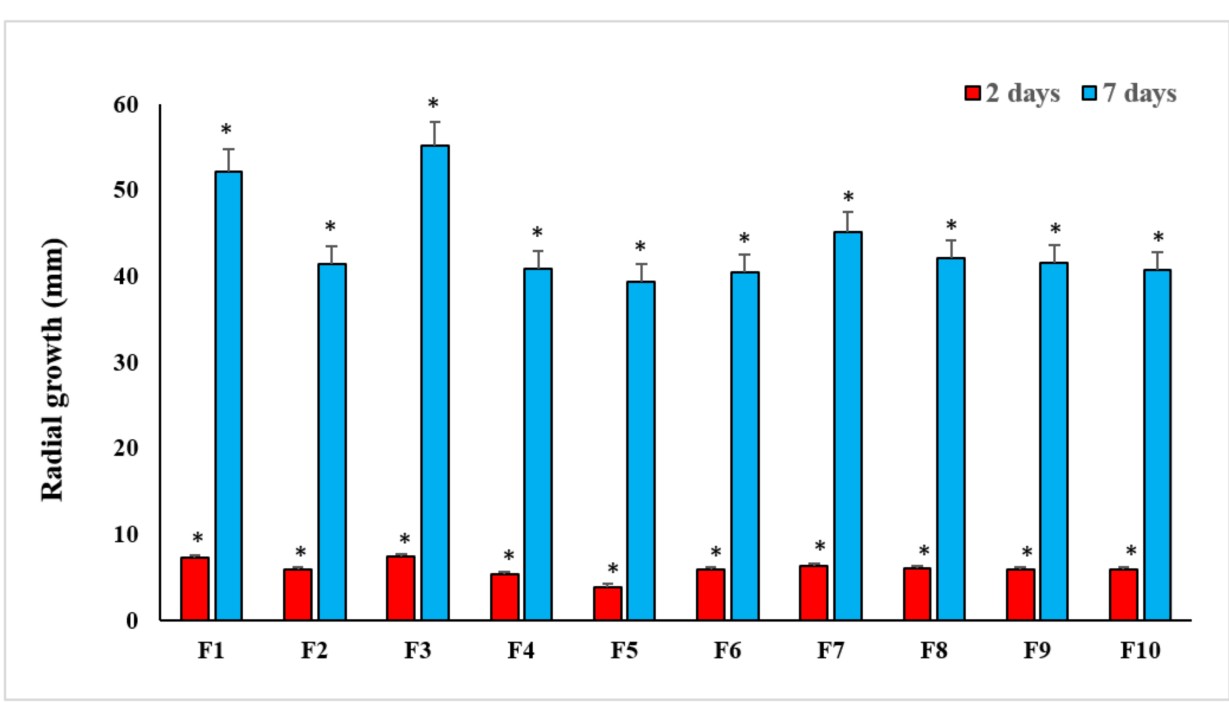

**Figure 1.** Radial growth of FOL isolates growing on a PDA medium at 25 °C (highly significant = * $p < 0.05$, $n = 3$).

### 3.3. Molecular Characterization of Fusarium Isolates

PCR amplification of the ITS area from 10 FOL isolates yielded an outcome of around 100 bp. The band size of isolate no. 4 changed significantly more than that of the other isolates (Figure 2). Furthermore, the nucleotide BLAST Program Fund performed a sequence similarity search for 11 isolates from the National Centre for Biotechnology Information (NCBI), Bethesda, Maryland, USA. Moreover, all of the isolates were recognized as *Fusarium oxysporum* f. sp. *lycopersici* (FOL). Five FOL isolates (F2, F3, F4, F7, and F8) had a 99.85% nucleotide resemblance to GenBank clone 105 with accession number MTCC8609. Three isolates (F5, F6, and F9) exhibited 99.50% nucleotide likeness to a GenBank clone 170 with accession number MTCC9912. The last two isolates (F1 and F10) showed a 100% nucleotide match to GenBank clone 42 with accession number MTCC8611 (Table 1).

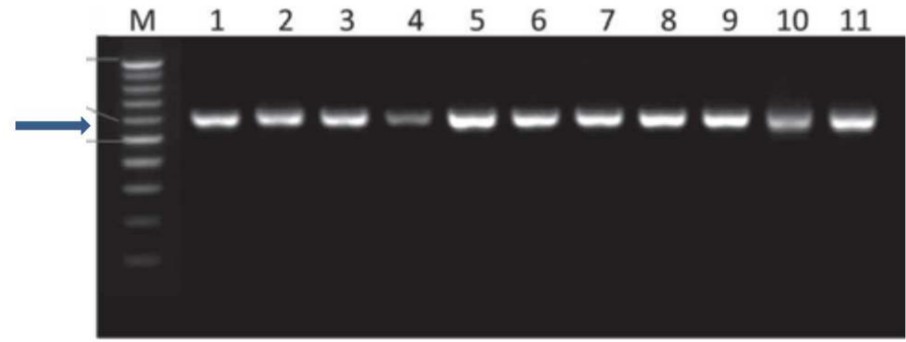

**Figure 2.** Internal transcribed spacer (ITS) regions of *Fusarium oxysporum lycopersici* isolates were amplified by polymerase chain reaction using ITS1 and ITS4 primers and the products were separated by agarose gel electrophoresis. M, 100 bp ladder; lane 1, F1 isolate; lane 2, F2 isolate; lane 3, F3 isolate; lane 4, F4 isolate; lane 5, F5 isolate; lane 6, F6 isolate; lane 7, F7 isolate; lane 8, F8 isolate; lane 9, F9 isolate; lane 10, F10 isolate; lane 11, no template control.

**Table 1.** Molecular variability of FOL isolates.

| Groups | Isolate Code | Accession Number | Closest Match | Similarity to GenBank Accessions | Frequency (%) |
|---|---|---|---|---|---|
| 1 | F2, F3, F4 F7, F8 | MTCC8609 | Clone 108 | 99.85% | 14.18 |
| 2 | F5, F6, F9 | MTCC9912 | Clone 180 | 99.50% | 40.72 |
| 3 | F1, F10 | MTCC8611 | Clone 108 | 100% | 25.10 |

### 3.4. Pathogenicity Testing

Tomato plants diseased with all FOL isolates showed wilting symptoms of varying severity. Data were obtained to confirm that F1 and F3 were very aggressive and produced the greatest severity of the disease. Isolates F2–F10, on the other hand, showed the least severe disease in tomato plants. Consequently, based on these data, isolate F1 was used for ongoing trials. Figure 3 shows a tomato plant infected with FOL (isolated strain F1) under greenhouse conditions.

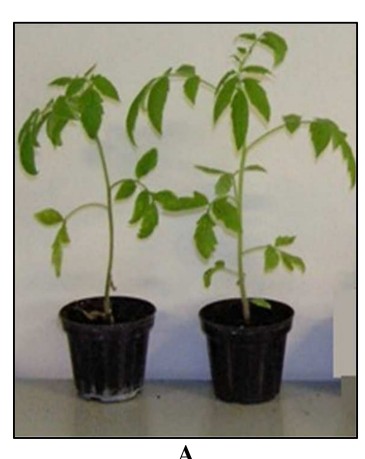

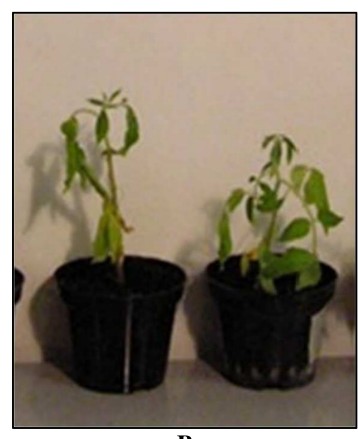

**A**

**B**

**Figure 3.** Pathogenicity test of tomato plant by FOL (Isolate F1) under greenhouse conditions. (**A**) Healthy tomato plants (control). (**B**) Infected tomato plants.

### 3.5. Effect of Algal Extracts on Linear Mycelial Growth of FOL In Vitro

The antifungal action of macroalgae extracts from *S. dentifolium*, *G. Compressa*, and *U. lactuca*, with chloroform, acetone, methanol, and water against FOL was calculated. As

shown in Figure 4A–C, the radial growth of FOL was significantly decreased at various levels. Methanol extracts of the three macroalgae showed the highest inhibition of FOL mycelial growth followed by chloroform compared to the control (90.0 mm). In addition, FOL radial mycelial growth was completely inhibited by the methanol extract of *S. dentifolium*. Figure 5 showed the antifungal activity of microalgae methanol extracts against FOL at different concentrations.

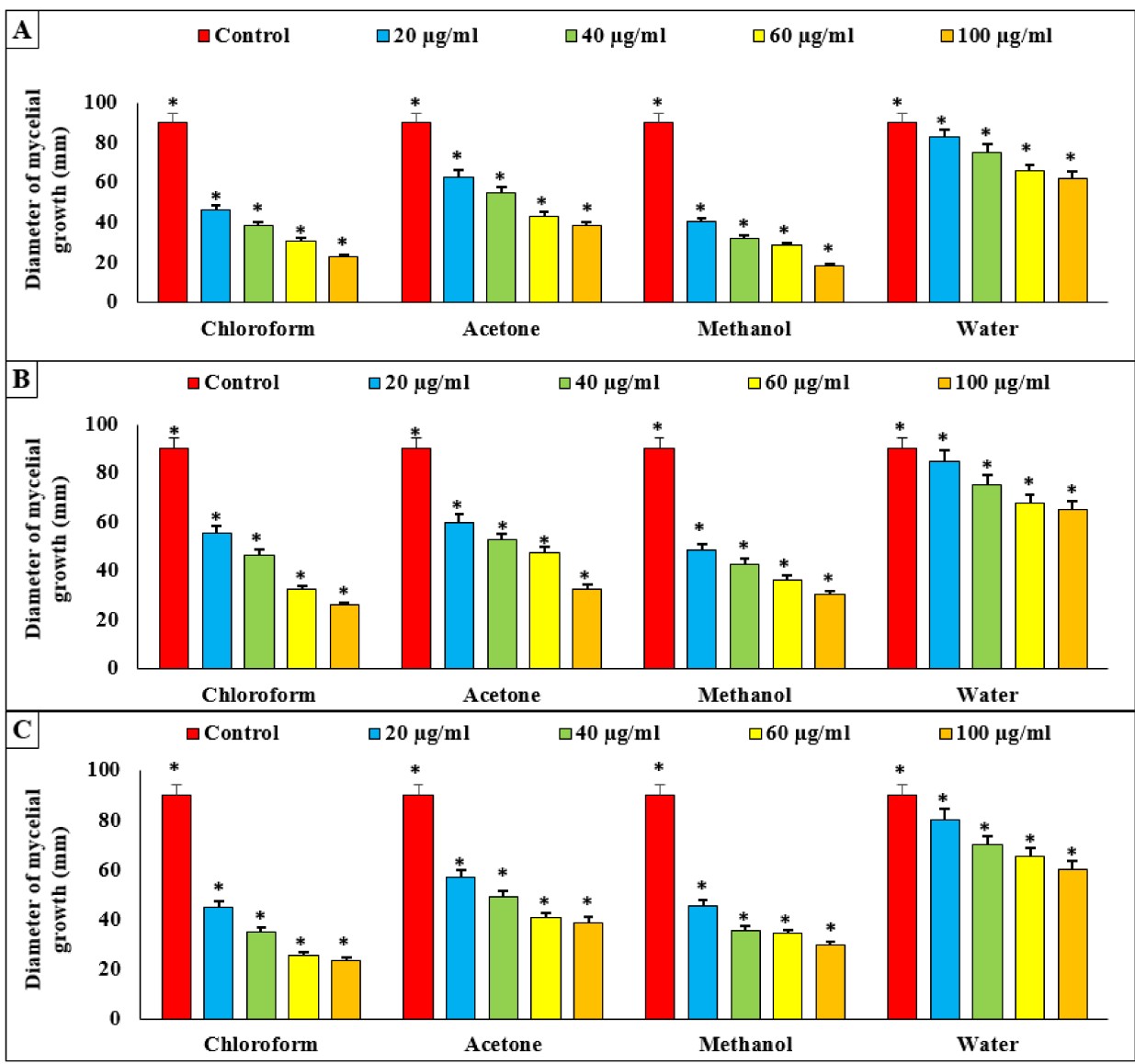

**Figure 4.** Effect of *S. dentifolium*; (**A**), *G. compressa*; (**B**) and *U. lactuca*; (**C**) extracts of different solvents on the mycelial fungal growth of FOL on PDA medium (highly significant = * $p < 0.05$, $n = 3$).

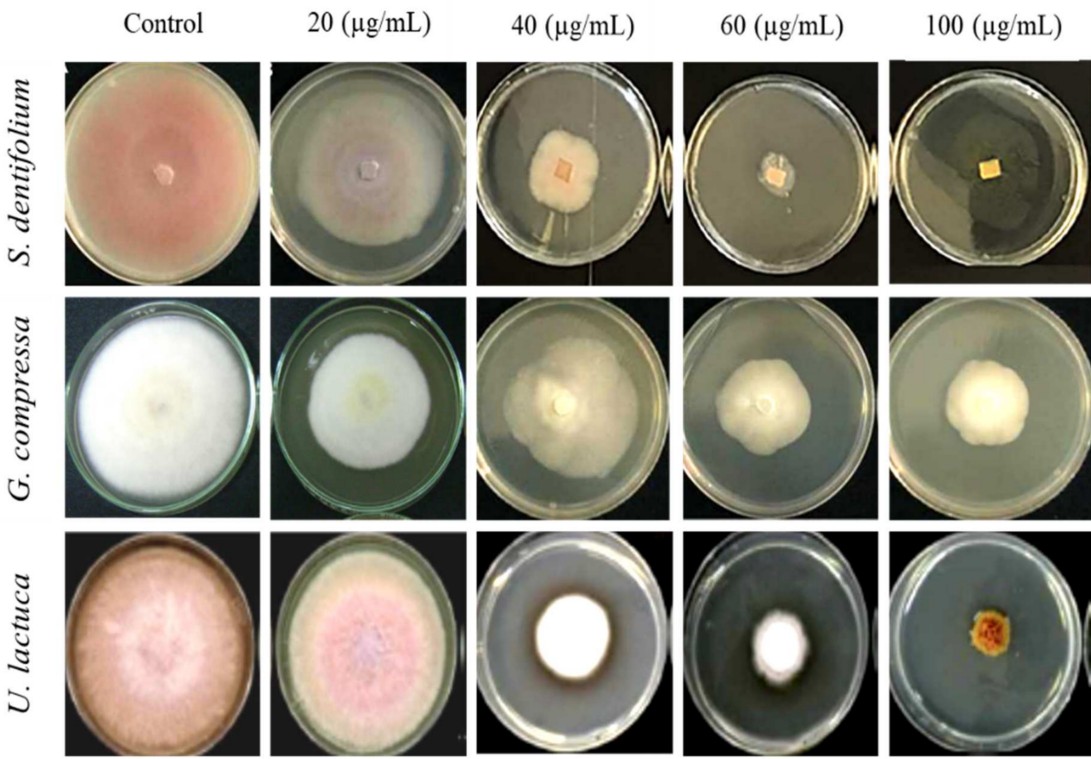

**Figure 5.** Antifungal activity of microalgae methanol extracts against FOL at different concentrations.

*3.6. Total Phenolic Content*

The Folin–Ciocalteu method revealed the phenol content in the different solvent extracts of the tested macroalgae and was reported in mg/g gallic acid as presented in Figure 6. Accordingly, the methanol extracts of the three macroalgae had the highest levels of total phenols; these extracts were recorded at 620.13, 595.22, and 580.87 for *S. dentifolium*, *G. compressa*, and *U. lactuca*, respectively. In addition, the extract of *S. dentifolium* presented the greatest phenol content of all solvents tested.

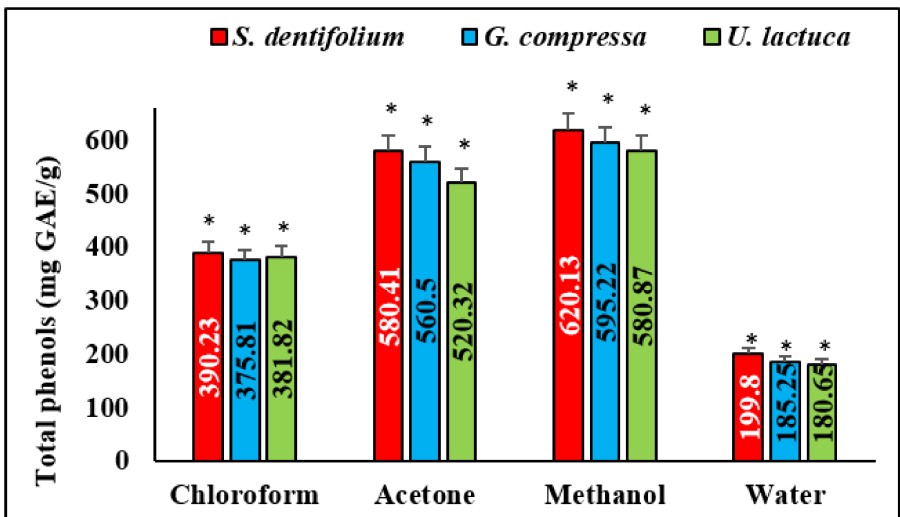

**Figure 6.** Total phenolic content of *S. dentifolium*, *U. lactuca*, and *G. compressa* extracts. The data are expressed as mg of gallic acid per gm dry extract (highly significant = * $p < 0.05$, $n = 3$).

### 3.7. Quantitative Phenolic Compounds Using HPLC

The phenolic composites in the extracts distinguished in the methanol extracts of the three algae were enumerated using HPLC and are presented in Figures 7–9. As evident from the data in Figure 7, phloroglucinol was highest (7.34 mg/g DW) in *S. dentifolium*, while catechin had the smallest value (0.82 mg/g DW). Gallic acid was prominent (6.89 mg/g DW) in *U. lactuca*, while caffeic acid had the lowest value (0.80 mg/g DW) (Figure 8). Furthermore, *G. compressa* had the highest quantity of vanillic acid (7.28 mg/g DW), and the lowest quantity of *p*-coumaric (0.82 mg/g DW) (Figure 9).

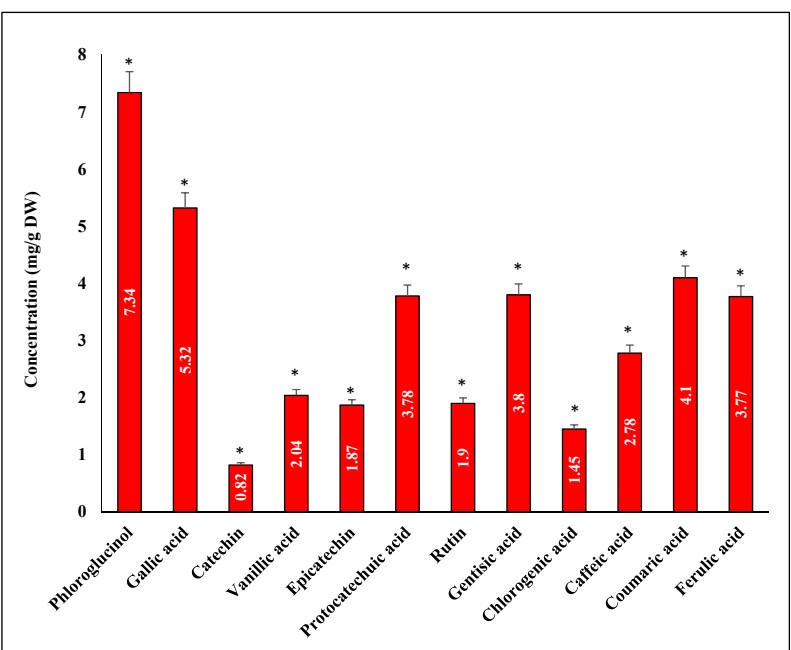

**Figure 7.** HPLC profile of phenolic compounds in methanol extract of *S. dentifolium* (highly significant = * $p < 0.05$, $n = 3$).

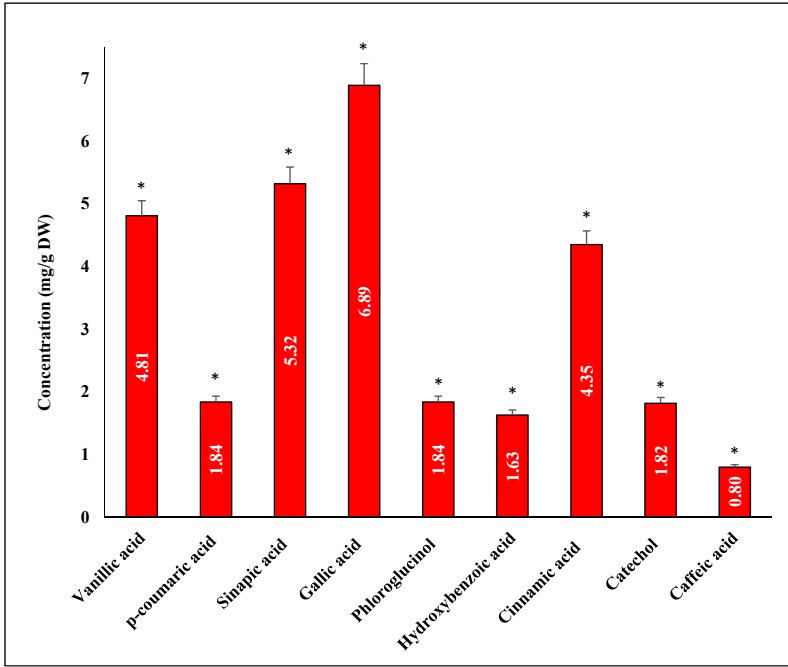

**Figure 8.** HPLC profile of phenolic compounds in methanol extract of *U. lactuca* (highly significant = * $p < 0.05$, $n = 3$).

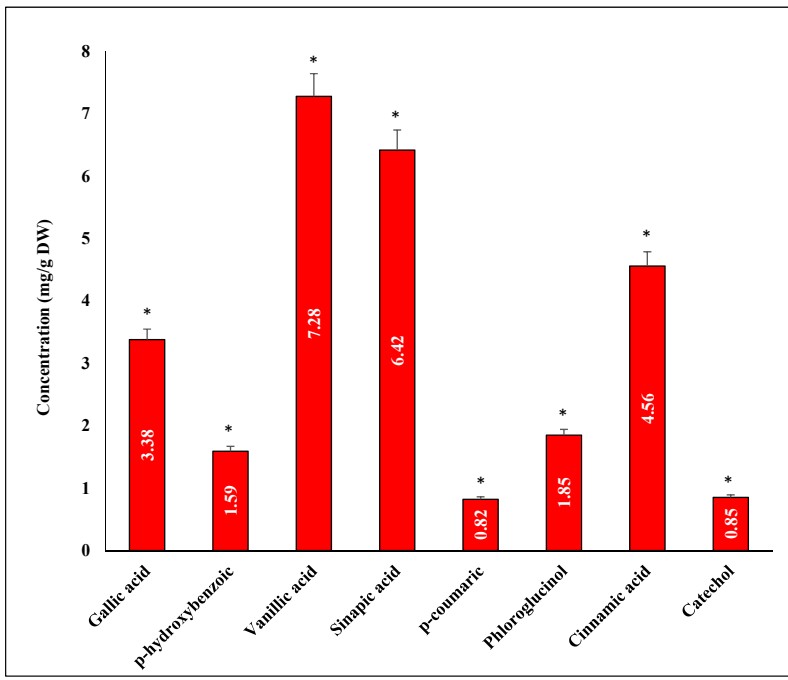

**Figure 9.** HPLC profile of phenolic compounds in methanol extract of *G. compressa* (highly significant = * $p < 0.05$, $n = 3$).

### 3.8. Scanning Electron Microscopy

The hyphae of FOL treated with the crude extracts of *S. dentifolium*, *G. compressa*, and *U. lactuca* (100 μg/mL) were examined by SEM to provide a well-defined picture of how these extracts might alter the hyphae of the pathogen. The effects showed a clear difference between the extract-treated and extract-untreated hyphae. The untreated FLO mycelium was well developed, bulging with smooth walls and prominent septa (Figure 10A). Conversely, mycelium treated with the methanol extract of the macroalgae displayed plasmolysis and crushing. Most hyphae seemed to be dead, especially in *S. dentifolium* and *G. compressa* extracts, but the septa were still prominent (Figure 10B–D).

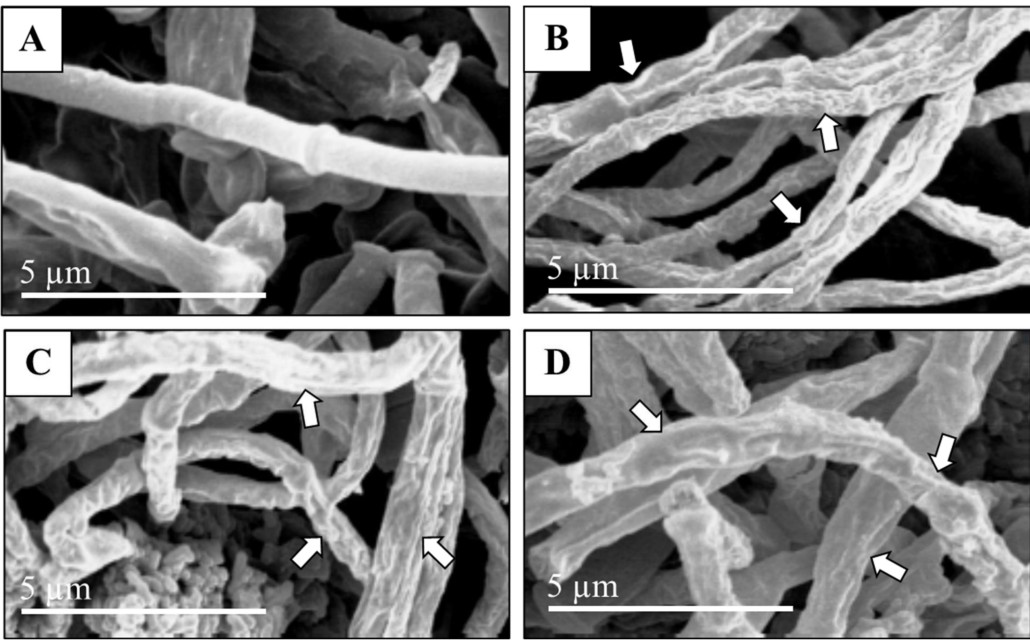

**Figure 10.** Scanning electron micrographs of FOL treated with *U. lactuca*, *S. dentifolium*, and *G. compressa*

extracts (100 µg/mL). The untreated mycelia are well developed and inflated with smooth walls (**A**). The treated mycelia showed plasmolyzed, distorted, squashed, and collapsed hyphae and some were completely dead (indicated by arrows); *U. lactuca* (**B**), *S. dentifolium* (**C**), and *G. compressa* (**D**). Scale bar = 5.0 µm.

### 3.9. Effect of Algal Powder of S. dentifolium on Disease Severity and Fruit Yield

The disease severity (percentage) of Fusarium wilt on tomato plants in glasshouses after treatment with *S. dentifolium* powder (10%) is presented in Figure 11. The data showed a significant decrease in the disease severity of the pathogen of 40.8% after treatment with the algal powder. In infected plants treated with fungicides, the disease severity diminished by 38.6%. In contrast, when the pathogen was treated with a combination of fungicide and algal powder, the disease severity was lowered by 56.3%. In general, fungicides used in combination with algal powder were less harsh than fungicides or algal powder used separately. Compared to the control value, the pathogen caused a dramatic decrease in the yield (450.3 g) of the tomato plant. Infected plants treated with algae powder had their tomato yield increased by 25.8%, while treatment with fungicides alone increased the yield by 20.5%. Plants with algal powder and fungicides had yields that increased by 32.3% compared to the two treatments described above (Figure 12A,B).

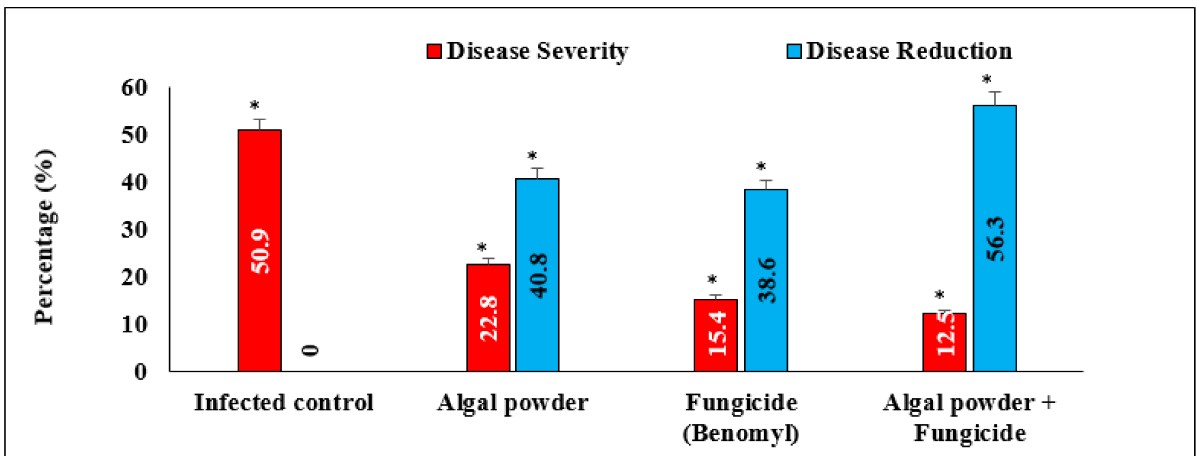

**Figure 11.** Disease severity of *Fusarium* wilt (percentage) in tomatoes treated with algal powder of *S. dentifolium* (at 10% concentration) (highly significant = * $p < 0.05$, $n = 3$).

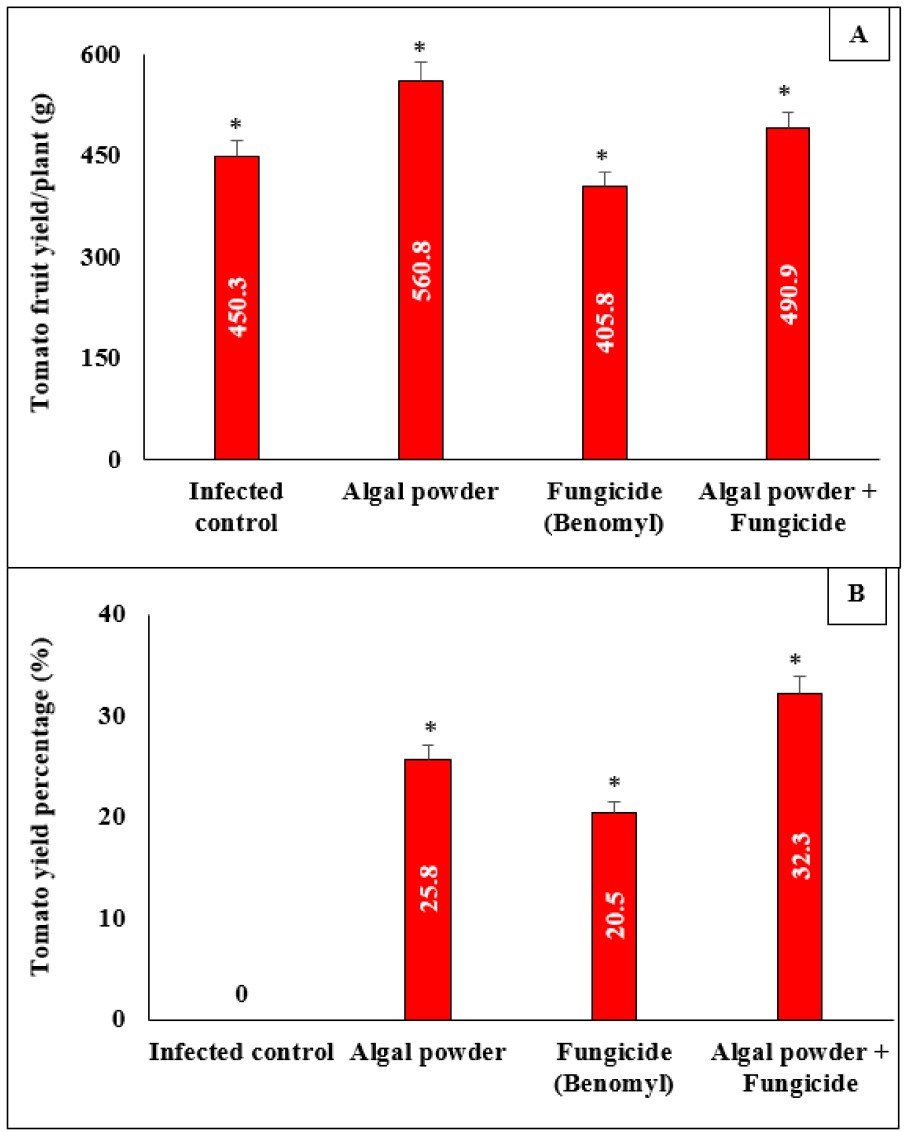

**Figure 12.** Influence of algal powder of *S. dentifolium* (10%) on tomato fruit yield. (**A**): tomato fruit yield/plant (g). (**B**): tomato yield percentage (highly significant = * $p < 0.05$, $n = 3$).

## 4. Discussion

This study screened crude extracts of three marine macroalgae related to Phaeophyta (*S. dentifolium*), Rhodophyta (*G. compressa*), and Chlorophyta (*U. lactuca*) for an antifungal activity for controlling the Fusarium wilt of tomatoes caused by FOL. A tomato Fusarium wilt is caused by three races of FOL. Races 1 and 2 are spread globally, while race 3 has a narrow geographic dispersal [27].

Many researchers in different countries have attempted to characterize the morphology and molecular properties of FOL [28–30]. In addition, the molecular studies in this study established the morphological properties of fungal isolates with supposed FOL. Therefore, while the morphological properties offered an outstanding means of fungal species identification, it was not possible to precisely identify isolates at the species stage. Others [31] found that the morphological category of fungal species lacked precision but was significant for grouping fungal isolates and facilitating validation with advanced approaches. In addition, morphological features such as colony color, texture, conidia size, and shape were used to distinguish Fusarium species [32]. The current investigation showed important morphological volatility inside FOL isolates.

Numerous investigators [33,34] have described the excessive genetic multiplicity of FOL. McDonald and Linde [35] suggested that genetic disparity might occur between the isolates developed from various lesions of the same leaflet. Moreover, as stated by Nirmaladevi et al. [36], genotypic dissimilarity in FOL is created by the capacity of its mycelia to make connections within hyphal merging that improve the dispersal of water, nutrients, and signaling molecules in the colony. In addition, genetic variety is similarly delivered by modifications, choice, and gene flow [37] heterokaryosis arises from hyphal anastomosis, recombination, and the progress of the pathogen over extended regions [38].

There are many accounts of bioactive composites resulting from macroalgae as a promising supply of biochemical and medical characteristics comprising antibacterial [39], antifungal [40], antiviral [41], antioxidant [42], and anti-inflammation [43] activities. Furthermore, the percentage of fungal mycelia decline in this study emphasizes the presence of bioactive compounds in the extracts of the macroalgae tested, which can be resolvable in solvents, and may be related to the high concentration in organic extracts against microorganisms [44]. Many researchers described the antifungal activity of some macroalgae against FOL in vivo and in vitro [45,46].

In the present study, chloroform, acetone, methanol, and water were used to extract the macroalgae tested. This screening indicated that methanol extract was the most effective and displayed a wide range of inhibitory effects on the fungal pathogen. Many investigators [47] stated that methanol extract had the highest antifungal activity and chloroform and petroleum ether had the lowest. In addition, organic solvents extract bioactive compounds from algae that are larger than water. Our findings are in line with earlier reports that stated that the antifungal activity of organic solvent extracts from algae was greater than that of water extracts [48]. This study reveals that pathogenic fungi respond differently to large algae extracts. The effect depends on the solvent used, the type of algae tested, and the environment in which the algae grow. In addition, several authors have come to different conclusions about the best solvents for the extraction of biologically active compounds. This study indicated that *S. dentifolium* extract completely inhibited the growth of FOL. We believe this is because *S. dentifolium* extract had the highest total phenol content of the algae tested. Phenols and flavonoids can play important roles in inhibiting the growth of some microorganisms [49].

Crude methanol extracts of the three macroalgae were analyzed by HPLC to identify the most important phenolic constituents that could play a part in controlling the fungal pathogen tested. The data presented here revealed the presence of many phenolic composites in the extracts of different macroalgae. In this study, we have confirmed the presence of large quantities of certain effective composites, e.g., phloroglucinol, gallic acid, and vanillic acid in *S. dentifolium*, *U. lactuca*, and *G. compressa*, respectively. *S. dentifolium* showed high amounts of phloroglucinol which may be a reason for the reduction in fungal growth. The prominent biological action of brown algae is frequently correlated with the occurrence of potent, non-lethal biological antioxidants. The antioxidant influence may be related to polyphenols and mainly phlorotannins, which are polymers of phloroglucinol [50]. These compounds are thought to be located only in brown seaweeds [51]. The antioxidant action of phlorotannins, obtained from diverse brown macroalgae species was confirmed in the laboratory [52]. In addition, high amounts of gallic and vanillic acids which were present in the extracts of *U. lactucae* and *G. compressa*, respectively, could be related to the fungistatic action. Gallic acid exhibited strong antifungal activity against *Fusarium solani* [53], and *Alternaria solani* [54]. Moreover, vanillic acid has been described as an antifungal compound against *Botrytis cinerea* [55].

Phenolic complexes are aromatic benzene rings with one or more hydroxyl groups produced by plants primarily to protect them from stress [56]. In addition, phenols are a major unit of secondary metabolites that are broadly dispersed in most macroalgae and that play a vital part in plant opposition to many diseases, acting as defenses against fungal pathogens and fungicides [57]. As a result, phenolic constituents have been proposed as an effective alternative to the chemical control of phytopathogens in crops. The sensitive

oxygen species that are produced by living organisms are considered to be an essential part of metabolism; they are very reactive and can produce dysfunction and cytotoxicity in cells. Polyphenols provide hydrogen to free radicals and can generate non-reactive radicals. In the current study, *S. dentifolium* contains quite a lot of polyphenols, while *U. lactuca* had the lowest content of these compounds. These results were consistent with those of Karou et al. [58].

The ethanolic extract from *Sargassum myricocystum* (brown alga) had a significant antifungal effect on *Colletotrichum falcatum* [59]. In addition, the presence of phenolic combinations, alkaloids, carbohydrates, and proteins, in a methanol extract of *Sargassum johnstonii* led to significant antifungal bioactivity [60]. Moreover, Ambika and Sujatha [61] reported that the methanol extract from *Turbinaria conoides* (brown algae) had an antifungal effect on *Pythium aphanidermatum*. The efficacy of bioactive compounds in macroalgae against FOL is based on the resistance induction mechanism in treated plants [62] or by inhibiting the growth of pathogen mycelium to cause infection in the early stages of growth [63].

The different antifungal activities of different solvent extracts against the fungal pathogen tested may be due to the different polarities of the extraction solvents. Therefore, the antifungal activity of the methanol extract in this study may be due to the abundant extract containing many phenolic components. Methanol is an excellent solvent for extracting many polar and some non-polar compounds. Phenols, glycosides, coumarins, sesquiterpenes, and alkaloids extracted with methanol have the potential and are expected to exert antifungal effects [64].

In addition, methanol extracts from a variety of macroalgae were used to examine FOL-treated hyphae. The results showed a clear difference between the treated and untreated mycelia. Mycelium treated at a dilution of 100 µg/mL showed significant changes such as plasmolysis and distortion, compared to controls. After the treatment of the fungal pathogen by *S. dentifolium* and *G. compressa* extracts, mycelia exhibited empty, disintegrated, and completely dead hyphae. Many researchers have explained how the bioactive natural substances found in plants and algae work. The antifungal action of such extracts can be ascribed to cell wall attack and hyphal cytoplasmic contraction, and ultimately to mycelial death [65]. Such modifications are triggered by the interference of macroalgae extract components with enzymatic reactions of wall synthesis that affect fungal morphogenesis and growth. Phenolic compounds in the extract have been shown to disrupt cell integrity, thereby inhibiting respiratory and ion transport processes [66].

In the greenhouse trial, the disease severity of Fusarium wilt on tomato plants was reduced after treatment with *S. dentifolium* powder. However, tomato yield was increased when they were treated with a similar powder. These outcomes are in agreement with the results of some authors [47], who explored the influences of macroalgae extract prepared from brown algae on the growth and productivity of crop plants. The application of seaweed extracts to the soil in which tomato plants were growing intensified the plant yield and quality. Ali et al. [67] stated that extracts created from the brown macroalga, *Ascophyllum nodosum* led to more flower bunches and fruit, higher shoot and root dry weights, and greater overall yields in the tomatoes grown in glasshouses and in fields.

## 5. Conclusions

This investigation supports the potential use of macroalgae extracts as a supplier of antifungal compounds and could be the foundation for a future study examining the use of algae as a new resource against phytopathogenic fungi. Finally, large marine algae are the basis of bioactive components and have possible applications in the control of fungal pathogens in the fields of medicine, pharmaceuticals, and agriculture. This can facilitate the use of natural products derived from macroalgae as an alternative to chemical fungicides to protect crops from fungal plant pathogens.

**Author Contributions:** Conceptualization, Y.S.M., S.A.A. (Saad A. Alamri), M.A.T., Z.A.B., S.A.A. (Sulaiman A. Alrumman); data curation, M.H.; formal analysis, M.H., M.A.T., Z.A.B.; investigation,

Y.S.M., S.A.A. (Saad A. Alamri), S.A.A. (Sulaiman A. Alrumman), Y.S.M.; methodology, M.H., Y.S.M., S.A.A. (Saad A. Alamri), S.A.A. (Sulaiman A. Alrumman), M.A.T., Z.A.B.; software, S.A.A. (Sulaiman A. Alrumman); writing—original draft, Y.S.M., M.A.T., Z.A.B., S.A.A. (Saad A. Alamri). All authors have read and agreed to the published version of the manuscript.

**Funding:** This research was funded by the Deputyship for Research and Innovation, Ministry of Education in Saudi Arabia (project number IFP-KKU-2020/2).

**Institutional Review Board Statement:** Not applicable.

**Informed Consent Statement:** Not applicable.

**Data Availability Statement:** Not applicable.

**Acknowledgments:** The authors extend their appreciation to the Deputyship for Research and Innovation, Ministry of Education (Saudi Arabia) for funding this research work, project number IFP-KKU-2020/2.

**Conflicts of Interest:** The authors declare no conflict of interest.

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
