# Peer review of "In Vitro and In Vivo Biocontrol of Tomato Fusarium Wilt by Extracts from Brown, Red, and Green Macroalgae"

_agriculture, doi:10.3390/agriculture12030345_

Round 1

Reviewer 1 Report

This manuscript provides interesting results, which could be potentially applied to disease control as a sustainable approach to reducing harmful chemical treatments.

However, there are some parts that need to be further improved for publication. The English writing for some parts, such as line 257-266 need to be improved.

For the method part, by checking with the Ithenticate, it showed some self-plagiarism and some of them did not provide the related citation ( 2.2-2.9.). I could understand that some method parts are difficult to change, but still many parts need further improvement.  Also, there is confusion about disease incidence and disease severity. The severity is the level of the disease, which needs the rating of the disease symptom level. I did not see any related description there. Therefore, please correct your term accordingly. If you did a rating analysis, then please add that part.

For the result part, please add significant analysis for your data analysis and update the related figures and legends. Even there is no difference, still, need to put the p value in the figure legends.

Author Response

Please see the cover letter concerning author response

Reviewer 2 Report

This manuscript firstly identified by morphological and molecular methods the pathogen caused by wilt disease of tomato, then investigated and compared the effect of the extracts from three macroalgae (Sargassum dentifolium belongs to Phaeophyta, Gracilaria compressa belongs to Rhodophyta, and Ulva lactuca belongs to Chlorophyta) against Fusarium oxysporum f. sp. Lycopersici, finally, analyzed the inhibitory efficiency on Fusarium oxysporum was due to high total phenolic content, and verified the combination of algal powder of Sargassum dentifolium and fungicide was the best substitution of chemical fungicide in field. The topic is of some interest. However, in order to improve the quality of the manuscript, some suggestions have been made.

  1. In “Introduction” part, the authors describe Macroalgae biocontrol effect, however, the part is a bit much, please rewrite this part of Line 52-92.
  2. In “material and method” part, the author identified the Fusarium pathogen caused tomatoes wilt just used the two primers of “ITS1” and “ITS4”, why not use “TEF”? please explain?
  3. In result part, the author identified ten Fusarium isolates with different characteristics of culture and morphology, but they are the same Fusarium oxysporum, how to explain?
  4. In result part, Line 329-333, from Figure 4 A, B, C, methanol extract showed the highest, followed by chloroform, but the author thought following by acetone?
  5. In figure 5, the picture is not clear, please change some pictures.
  6. In figure, no data about lactuca, please delete it.
  7. In Figure 8, 9, 10, the author determined profile of phenolic compounds in methanol extract of dentifolium, U. lactuca, G. compressa by HPLC, but in “discussion” part, the author did not analyze the difference between these phenolic compounds to influence the inhibitory effect.
  8. In “Result” part, about the result of SEM, “FLO mycelium is well-developed, bulging with smooth walls and prominent septa”. From SEM, we can see smooth walls and prominent septa?
  9. Please mark the scale in Figure 10.
  10. Please check Line 172-173 of primer of ITS1 and ITS4.
  11. During HPLC determination, Line 228, the insertion volume was 20 mL, is it correct?

Author Response

See cover letter concerning author response

Reviewer 3 Report

This paper presents a work in which some algae extracts have been investigated for their potential to control Fusarium wilt of tomato caused by the fungus Fusarium oxysporum f.sp. lycopersici (FOL). The topic of this research is interesting because Fusarium wilt of tomato is an important disease of this crop in the main tomato-producing countries worldwide. Nevertheless, the research presented in this manuscript and the methodology used contain some weaknesses. Some of my main concerns are listed below:

-The sections 2.1, 2.2, and 2.3 should be simplified. It is not needed to explain in detail how the isolates used in the experiments were obtained, the pathogenicity test on tomato plants, and the morphological study. The authors just need to state that they used well characterized FOL isolates obtained from diseased tomato plants. But, on the contrary the molecular characterization is not correct. Currently, molecular identification of Fusarium strains is done by sequencing the Elongation Factor (EF) gene region, not ITS. Moreover, the authors do not provide information about to which race or races belong the isolates used. This is relevant, and also to know if the tomato cultivar used for the experiments has resistance genes introduced against FOL or not.

-The experiments to assess the effect of algae extracts against mycelial growth use PDA Petri dishes amended with different volumes of algae extracts, but the controls are just PDA+distilled water. In my opinion the controls are incomplete. There must be controls in which PDA plates are amended with each one of the solvents used to prepare the algae extracts alone: chloroform, acetone and methanol. If not, how can the authors determine (or separate) the potential negative (toxic) effects that these compounds could have themselves against mycelial growth of FOL strains? Perhaps the effects observed in the in vitro experiment are due to the toxicity of these compounds and not due to algae composition. This can also have affected the aspect of hyphae when observed under scanning electron microscopy.

-Why the authors only explored the effect of algae extracts against mycelial growth and not against conidia germination? When a fungus such as FOL easily produces conidia, it is important to evaluate the effect of fungicides and other substances against both mycelial growth and spore germination. In this case, the effect on conidia germination is especially relevant, because FOL is a soilborne pathogen in which root infection occurs after conidia or chlamydospore germination.

-In the pathogenicity tests soil was infested with an inoculum carried by sorghum grains. There is no description about how the sorghum grains were infested; for instance, if they were previously autoclaved or not, time and conditions of incubation of this inoculum, etc. Moreover, with this type of inoculum there is not an exact quantification of the FOL conidia introduced into the pots. Most of the inoculations performed with FOL are performed using spore (conidia) suspensions of this fungus, in which it is possible to measure the amount of conidia per ml. Thus, this facilitates standardization of the same amount of inoculum in each infested pot. Moreover, this experiment was not repeated. Why in this pathogenicity test the combination Algal powder (S. dentifolium 10%)+fungicide was included as shown in Figure 11? This combination is not included in the description of the pathogenicity test (2.10) and, in my opinion; it is not an interesting option if the context of this research is organic tomato production.

- The manuscript is poorly written and needs additional revision. Some examples of this are:

In the Keywords, scientific names are not in italics. This is a mistake.

In the text the abbreviation of Fusarium oxysporum f.sp. lycopersici is indicated as FOL but also many times as FLO. This is a mistake.

In Figure 5 the pictures of the control and 20µg/ml are the same for both S. dentifolium and G. compressa. This is a mistake.

In Figure 6, the foot note includes S. dentifolium, U. lactuca and G. compressa, but the figure deals only about S. dentifolium and G. compressa.

In Figure 12 it is not indicated what letters A and B mean.

Author Response

see cover letter concerning authors response

Round 2

Reviewer 3 Report

The authors of this manuscript have done an effort to improve the quality of the manuscript and also to respond to all my comments. In my opinion it still contains some scientific weaknesses, but I think that it deserves one additional revision. In this sense, I would like to suggest to ask the authors to resubmit it again as a clean version, without all changes indicated using the track-changes of the word system. In the current form the manuscript is very confusing and difficult to revise and, thus, it is also very difficult to reach a definitive decision about an overall recommendation. 

I would like to suggest the authors to perform a very detailed revision of the text, tables and figures of this clean version before resubmitting it, in order to remove any mistakes, and also to revise the correctness of the English language.

Author Response

Dear Reviewer;

Thank you for your valuable comments

I revised the manuscript carefully and submitted it after the corrections you requested

I submitted it as a clean revised version and all tables, and figures are inserted in their places

Concerning the English language, I corrected all mistakes after editing by MDPI

Please see submitted clean revised version
